# Clinical Improvement in Depression and Cognitive Deficit Following Electroconvulsive Therapy

**DOI:** 10.3390/diagnostics13091585

**Published:** 2023-04-28

**Authors:** Ahmad Mus’ab Ahmad Hariza, Mohd Heikal Mohd Yunus, Jaya Kumar Murthy, Suzaily Wahab

**Affiliations:** 1Department of Physiology, Faculty of Medicine, UKM Medical Centre, Jalan Yaacob Latiff, Bandar Tun Razak, Kuala Lumpur 56000, Malaysia; 2Department of Psychiatry, Faculty of Medicine, UKM Medical Centre, Jalan Yaacob Latiff, Bandar Tun Razak, Kuala Lumpur 56000, Malaysia

**Keywords:** electroconvulsive therapy, cognitive deficit, hippocampus, neurogenesis, neuronal edema

## Abstract

Electroconvulsive therapy (ECT) is a long-standing treatment choice for disorders such as depression when pharmacological treatments have failed. However, a major drawback of ECT is its cognitive side effects. While numerous studies have investigated the therapeutic effects of ECT and its mechanism, much less research has been conducted regarding the mechanism behind the cognitive side effects of ECT. As both clinical remission and cognitive deficits occur after ECT, it is possible that both may share a common mechanism. This review highlights studies related to ECT as well as those investigating the mechanism of its outcomes. The process underlying these effects may lie within BDNF and NMDA signaling. Edema in the astrocytes may also be responsible for the adverse cognitive effects and is mediated by metabotropic glutamate receptor 5 and the protein Homer1a.

## 1. Introduction

Major depressive disorder (MDD) is a debilitating illness that affects one out of five people at some point in their lives [1]. MDD is associated with higher treatment costs across all age groups, especially when presented as a comorbidity [2]. While the DSM-IV defines MDD as the presence of a defined set of symptoms that persists for nearly 2 weeks [3], it also encompasses a number of subtypes based on severity, onset, and clinical features [1].

Electroconvulsive therapy (ECT) is a procedure that uses an electrical current to induce seizures in patients. The seizures are then followed by a clinical remission of the psychiatric symptoms. The seizures induced by ECT are typically generalized tonic–clonic (grand mal) seizures [4]. The tonic phase occurs first; clinically, this is presented as a stiffening of the body that lasts for a brief few seconds. It is subsequently followed by the clonic phase, which involves shaking or jerking of the body and lasts for tens of seconds [5]. Although ECT is used to induce seizures, it does not typically cause epilepsy in patients [6].

ECT has long been regarded as a safe and effective treatment option for clinical depression, especially in patients who are resistant to antidepressant medications [7]. When used in conjunction with antidepressants, it produces better results compared to pharmacotherapy management alone [8,9,10]. Other types of depressive disorders, such as suicidal depression, bipolar depression, psychotic depression, depression in the elderly, and postpartum depression, have benefited from ECT as well [11,12]. ECT is also indicated for bipolar disorders, treatment-resistant schizophrenia, catatonia, and neuroleptic malignant syndrome [13]. The administration of ECT alone or in combination with antidepressant medications is more effective in reducing depressive symptoms than antidepressant medications alone [14]. Continuing ECT with adjunct pharmacotherapy after an acute ECT course reduced the likelihood of relapse and recurrence of depression for up to a year [8]. According to a meta-analysis study, it is a more effective treatment for depression compared to repetitive transcranial magnetic stimulation, another non-pharmacological treatment for major depression [15].

The clinical benefits of ECT also come with some adverse effects, such as cognitive deficit [16]. Although the current trend in ECT practice is to reduce the cognitive side effects and maximize the therapeutic effects [11], the mechanism of ECT remains poorly understood. The interplay between neurogenesis and the inflammatory process is implied in the therapeutic mechanism [17,18,19], but little is known about their relation to the cognitive outcome.

The hippocampus is involved in memory and emotion regulation [20]. It can be anatomically divided into the hippocampus proper, which includes the cornu ammonis subfields (CA1 to CA3), and the dentate gyrus (DG), which includes a molecular layer, granule cell layer, and hilus (also known as CA4). Functionally, the hippocampus can be divided along the longitudinal axis into the anterior, middle, and posterior segments. The posterior hippocampus provides spatial learning, memory, and perceptual functions, while the middle segment is involved in cognitive functions [21,22]. The anterior hippocampus regulates the emotional and affective processes [23]. The hippocampal role in emotion regulation is due to its extensive connectivity with the amygdala and the prefrontal cortex (PFC). The PFC increases hippocampal activity during memory elaboration, with memories eliciting stronger effects carrying greater activity [24].

The hippocampus is central to the pathophysiology of depression. For example, chronic stress, which is the often-cited cause of depression, causes glucocorticoid dysregulation, resulting in hippocampal dysfunction [25]. Diminished monoamine neurotransmitters such as norepinephrine, serotonin, and dopamine in regions such as the hippocampus, cerebral cortex, and midbrain have also been implicated in the mechanism of depression [26]. The hippocampus is also a component of the brain’s default mode network and salience network. Depression symptoms are associated with hyperactivity in the default mode network and hypoactivity in the salience network [27,28,29,30]. While hippocampal changes are known to occur following ECT, little is known about their relationship to clinical and cognitive outcomes.

Numerous works in the literature have discussed the effects of ECT and the neurogenesis mechanism that underpins these effects, but very little has been written about the cognitive side effects and their mechanisms, especially given the current state of knowledge. In a recent commentary by Cavaleri and Bartoli, despite the various biomolecular mechanisms implied in the therapeutic and cognitive outcomes of ECT, much of what is known is still very little, due to the fact that most related studies are limited by their small sample size and methodological heterogeneity [31].

This review attempts to highlight the various changes caused by ECT on various brain structures, with an emphasis on the hippocampus, and to form a narrative as to how the changes are related to the therapeutic and cognitive effects of ECT and the potential mechanism underlying these effects. Literature searches were conducted by inputting relevant keywords into several different databases. Studies are limited to within 5 years, but older studies were included to support the narrative.

## 2. ECT Alters the Brain Structures

### 2.1. Structural Changes in the Hippocampus in Depression and Following ECT

The hippocampus is susceptible to dystrophic changes caused by depression, which is mediated by chronic stress. When measuring morning serum cortisol levels in MDD patients, it was discovered that higher levels of cortisol were associated with lower volumes of the hippocampal subfields, especially on the right side [32]. Children and adolescents also suffer from the adverse effects of early life stress, which may lead to depression. In a study measuring hair cortisol levels in children, higher hair cortisol levels were significantly associated with reduced volumes in the CA3 and DG, but not in the CA1 or subiculum [33]. Higher perceived stress in adolescents was associated with reduced total hippocampal volume and the left hippocampal volume [34]. Early life stress and greater depression severity in children were also linked to smaller hippocampal volumes [35,36,37]. Other inflammatory markers, such as IL-6 and CRP, were also elevated in MDD patients, but they had no association with changes in the hippocampal subfields [38]. In a preclinical study, rats subjected to chronic unpredictable restraint stress for 4 weeks had a smaller total hippocampal volume. This reflects volume reductions in all subfields in both the dorsal and ventral hippocampus [39].

ENIGMA meta-analyses of magnetic resonance imaging (MRI) studies on subcortical structures in MDD patients revealed significantly reduced hippocampal volumes in MDD patients, of which the effect was driven by recurrent MDD [40]. Other studies found MDD to be related to either left, right, or both hippocampal volume reductions [41,42], with a more substantial loss in the left hippocampus of MDD and bipolar disorder (BD) patients [43]. Variations in the findings may be attributed to factors such as the patients’ age or sex, the different spectra of depressive illnesses, or the duration of the illness. Within the hippocampal subfields, volume effects were confined to the principal subfields such as CA1-CA4, DG, and the subiculum, which were more pronounced on the left side and in recurrent depression [44]. While recurrent depression had a more marked reduction in the CA1 area, only the CA2-CA4 areas were reduced in the first episode of MDD [44], which may be due to a reduction in the bilateral parahippocampal gyrus [45]. A meta-analysis of voxel-based morphometry (VBM) studies revealed smaller gray matter volume in the left hippocampus in MDD and BD patients relative to the control group [43]. Adolescent BD patients with a history of suicide attempts had more GMV reduction in the bilateral hippocampus compared to adolescent MDD patients with a history of suicide attempts [46]. Despite being generally associated with volume reductions, the volume of the hippocampal tail was found to be larger in MDD patients compared to healthy controls. This correlated positively with clinical remission, implying that it could be used as a biomarker for antidepressant treatment [47].

ECT induces structural changes in the hippocampus. Longitudinal imaging studies in depressed patients receiving ECT treatment revealed a significant volumetric increase in the medial temporal lobe regions [48,49,50,51], where the hippocampus resides. ECT has been reported to significantly increase hippocampal volume [49,52]. T1 images obtained from the 3T MRI scanner revealed a significant volume increase in most of the hippocampal subfields, including both left and right DG [53], the right GCL and ML [54,55], the left GCL, the right CA3 and CA4, and the right subiculum [54], as well as the right hippocampal–amygdala transition area [55]. The volume increase in the right CA4 and DG distinguishes remitters from non-remitters [56], as do neuroplastic changes in the neural fiber connections between the hippocampus and other brain regions [57]. While ECT resulted in a significant volume increase in all subfields except the right fimbria [53], another study employing a more powerful MRI at 7T determined that significant volume increases occurred exclusively within the DG, while the other subfields were unaffected [58]. Although this study reaffirms that neuroplastic changes, specifically neurogenesis, occur solely in the DG as a result of ECT, other processes may be attributed to other subfields or regions, namely synaptogenesis or edema.

An interesting disparity was seen between the CA1 subfield, which is affected most by depression, and the DG, which receives a robust volume increase after ECT. As the CA1 is involved in the hippocampal consolidation and retrieval of episodic and autobiographical memory [59], and the DG is the site of adult hippocampal neurogenesis, the relationship between these sites in the context of depression and the ECT mechanism should be further explored.

### 2.2. Structural Changes in the Amygdala in Depression and Following ECT

Similar to the hippocampus, the amygdala is associated with emotional dysregulation, and has structural changes in depression [60]. According to a meta-analysis, patients with their first episode of MDD who presented with a higher Hamilton Depression Rating Scale (HDRS) score were more likely to have reduced GMV in the right amygdala [45]. An earlier meta-analysis revealed that an earlier age of onset of MDD was linked to a smaller amygdala volume [40]. However, there is evidence stating otherwise. Treatment-resistant depression (TRD) patients had higher GMV in both the right and left amygdala than non-TRD patients [61]. As this difference was not related to medication or bipolar or unipolar features, it may reflect a vulnerability to chronicity [61]. Increased right amygdala volume may occur first in the early stages of the disease, which then progresses to an increase in both as the illness’s chronicity increases. Another meta-analysis determined that depressed patients with comorbid anxiety showed significantly larger amygdala volumes [62]. This may be attributed to the amygdala being the site for the emotion processing of fear and anxiety [63]. Mice subjected to chronic restraint stress (CRS) experienced the persistent activation of projecting neurons from the basolateral amygdala to the ventral hippocampus, which is correlated with stress-induced anxiety-like behavior [64]. From this, it can be deduced that amygdala involvement in MDD is correlated with anxiety symptoms. The involvement of both the right and left amygdala may also differ with the illness’s chronicity, with only the right amygdala experiencing a volume increase in the early stages of the illness and then both sides increasing as the illness progresses.

As ECT influences morphometric changes in the medial temporal regions, other structures within these regions besides the hippocampus are also affected. The parahippocampus, insula, fusiform gyrus, and the amygdala are among these structures [48,49]. The amygdala, in particular, received similar volume increments together with the hippocampus [48,51,65], as both sites are associated with emotional processing and memory [51]. Within the amygdala, these changes were localized in the nuclei within the basolateral and centromedial amygdala and the corticoamygdaloid transition area [55,65]. In addition, VBM and resting state functional connectivity analysis revealed increased GMV in the left superficial amygdala, as well as enhanced connectivity between the left amygdala and the left fusiform area [66]. Similarly, a high-density EEG study revealed significantly increased outgoing functional connectivity in the right amygdala in patients with moderate to severe depression compared to healthy controls [67].

### 2.3. Structural Changes in Other Brain Regions in Depression and Following ECT

Multiple systematic reviews and meta-analyses have determined cortical thinning in MDD patients, including the bilateral orbitofrontal gyrus [68,69,70], the left calcarine fissure and lingual gyrus [70], the middle temporal gyrus [68,69], the gyrus rectus extending into the right striatum [45,68], the dorsolateral aspect of the right superior frontal gyrus, the medial aspect of the left superior frontal gyrus, and the left superior parietal gyrus [45]. Increased cortical thickness was also found in the anterior and posterior cingulate cortex, as well as the ventromedial PFC [68,69]. Another meta-analysis found that while both MDD and BD patients had lower GMV relative to healthy controls in areas such as the dorsomedial and ventromedial PFC, the anterior cingulate cortex (ACC), and bilateral insula, MDD patients had a more robust GMV reduction in the right dorsomedial PFC compared to the healthy controls. This is in addition to a reduction in the cerebellar, temporal, and parietal regions [43]. Both MDD and BD patients were discovered to have lower GMV in the left ACC when compared to healthy controls, and MDD patients had smaller GMV in the left superior frontal gyrus and the left ACC when compared to BD patients [71]. Besides the cortical changes, volume reductions were also found in the putamen, pallidum, and thalamus of MDD patients without comorbid anxiety [62].

Changes in the brain after ECT were also found in other brain regions besides the medial temporal region. One meta-analysis noted that volumetric increases after ECT occurred in the cortical and subcortical areas [72]. An imaging study showed a similar volumetric gray matter area increase in both general areas [73], implying a widespread effect on the brain. The specific areas that are noted to have a significant increase include the bilateral inferior temporal cortices, the supramarginal gyrus extending to the postcentral gyrus, the middle part of the cingulate gyrus, the right anterior cingulate gyrus and the right subgenual anterior cingulate gyrus, the left insula, and the left fusiform gyrus [48,49,74]. Other notable changes include increased volume in the left superior and inferior temporal gyrus, as well as hypergyrification in the left middle temporal gyrus [74,75]. There are also regions that are spared from any changes post-ECT, namely the brain stem and the bilateral cerebellar cortex [76].

### 2.4. Influence of Electrode Placement and Number of ECT Sessions on Brain Structural Changes

The magnitude and laterality of the structural changes in the brain are mediated by electrode placement. While bilateral electrode placement has been shown to increase the volume of certain hippocampal subfields on both sides [53,54], it was also reported by other studies to produce a larger volume increase in the right hippocampus and its subfields [48,56,77]. Bitemporal electrode placement was noted to differ from bifrontal placement in that it produces greater direct electrical stimulation of deep midline structures, the left and inferior temporal regions, the central and lateral sulci, and the ventral parts of the brain. It also affected a larger area of the left hemisphere, extending to the inferior frontal cortex, the entire left temporal lobe, and a large part of the left parietal lobe [74]. Studies employing RUL ECT produced a more consistent outcome in which the right hippocampus developed a more significant volumetric increase [49,50,55,75]. Structures deep within the brain or that are further away from the electrodes on the scalp may show a proportional decrease in ECT influence on brain matter [76]. As electric signals from the electrode travel across different tissues (skin, cerebrospinal fluid (CSF), and brain matter) with varying conductive properties, any structures further away from the electrode should receive fewer electrical signals compared to those nearer the electrodes, resulting in regional differences in volume changes [78]. Even though this may in part explain the greater volume changes with bilateral electrode placement as it covers more brain regions [79], another factor to consider is the relationship between individual brain structures and their own respective electric fields. A strong relationship was found within the left hippocampus and left amygdala [78], explaining why these structures changed the most with bilateral ECT. The influence of the electric field was supported by a recent study that determined that there is a stark difference in the electric fields between electrode placements. The highest electric fields for bilateral ECT were found in the white matter of both temporal lobes and the frontal lobes, whereas RUL ECT displayed the highest electric fields in the right hemisphere and the corpus callosum [80].

The number of ECT sessions was found to influence hippocampal volume change in a dose-dependent manner. This means that there is an increasing hippocampal volume that correlates with the number of ECTs administered over the course of an ECT treatment series [75]. This relationship holds true regardless of ECT parameters such as electrode placement and number of sessions, as well as the current, frequency, and duration of the electrical stimulation [75,81]. Another study, which only included subjects who received RUL ECT, found a dose-dependent effect between the strength of the electric field and hippocampal volume change [78]. The strength of the electric field is related to the number of ECT sessions and is also influenced by electrode placement, since RUL ECT produces a more robust volume increase that is lateralized to the right side [78]. It is worth noting that the dose-dependent relationship between the electric field and hippocampal neuroplasticity may be confined to a specific range between 30–40 V/m and 100 V/m [78]. This phenomenon, termed “neuroplasticity threshold,” means that a minimum electric field of 30–40 V/m is required for induction of neurogenesis, while any value above 100 V/m represents a less robust dose-response relationship and may be unrelated to further volumetric increase and antidepressant response [78]. Thus, there is a dose-response relationship between the number of ECT administrations and the increase in hippocampal volume, which is influenced by the electric field and the electrode placement modality.

### 2.5. The Relationship between Post-ECT Volume Changes and Clinical Outcome

Several studies have pointed out a significant association between post-ECT volumetric changes and improved clinical outcomes. Increased volumes of both the hippocampus and amygdala with ECT are related to improved depression symptoms [65,72,73]. Among the hippocampal subfields, a positive correlation with improved clinical outcome was found to be localized within the DG [58], although other subfields were also mentioned to have similar correlations, such as the CA3, CA4, and subiculum [54,56]. A positive correlation with improved symptoms was found to occur within almost all of the hippocampal subfields, except the right subiculum and fimbria, and the left presubiculum, parasubiculum, fimbria, and the hippocampal tail [53]. These findings supported the idea of hippocampal neuroplasticity as the mechanism necessary for improved clinical outcomes.

In contrast with the previous statement, there is also evidence that suggests that post-ECT volumetric changes may not be correlated with improved clinical outcome. A few meta-analyses determined that volume changes in the hippocampus after ECT had no correlation with reduced depression symptoms [52,82]. Similarly, one study found no link between an increase in left amygdala volume after ECT and an antidepressant response [78]. It seems that the volumetric increase of gray matter in the cortical and subcortical areas [79], especially in the temporal regions [50], had no association with any clinical improvement. Patients with greater treatment-related volume increases had poorer outcomes, though these effects were not significant after controlling the number of ECTs given to these patients [75]. These findings challenge the notion of hippocampal and amygdala volume increase following ECT as the underlying mechanism of clinical improvement. Additionally, when taking the influence of electrode placement and the electric field into account, stronger electric fields in the temporal lobes were associated with poorer clinical responses in bilateral ECT, while no differences in electric field distributions were found between RUL ECT responders and non-responders [80]. It could be inferred here that the greater electrical stimulation of bilateral ECT on the temporal lobe structures, particularly the left hippocampus and amygdala, may reach a certain threshold that impedes clinical remission. It could also explain why a greater increase in hippocampal volume post-ECT was associated with a poorer outcome [75].

### 2.6. The Relationship between Post-ECT Volume Changes and Cognitive Outcome

A growing body of evidence demonstrates that impaired cognitive performance occurs alongside a volume increase in the brain following ECT (Table 1). Enhanced cortical neuroplasticity after ECT has been linked to both clinical remission and impaired delayed memory, more specifically in the left inferior parietal gyrus and the right inferior temporal gyrus [83]. Increased total hippocampal volume and its subfields in depressed patients who underwent bitemporal ECT were shown to significantly correlate with a reduction in cognitive performance and worsening verbal memory immediately after completing their treatment [17,84]. However, after a 6-month follow-up, there was a reduction in the hippocampal subfields and improved cognitive performance. Improved delayed verbal memory was significantly associated with volumetric reduction in the right hippocampus and its subfields, namely the right DG, the right molecular layer, the right CA3, and the presubiculum, as well as the left subiculum [17]. Another study on RUL ECT noted that a larger absolute right hippocampal volume change is associated with less improvement in visual memory when assessed 6 months after completing the ECT course [85]. While this may indicate that absolute left hippocampal volume change was not significantly correlated with cognitive change [85], an earlier study explained that the greater clinical remission and cognitive deficit that occurred in bilateral ECT were due to greater direct electrical stimulation of the left hippocampus and inferior frontal cortex [74]. In a recent cohort study by Argyelan et al., it was reported that hippocampal enlargement following ECT is associated with poorer cognitive outcomes, regardless of diagnoses, electrode placements, and number of ECT sessions [86].

Another aspect worth mentioning is that hippocampal changes following ECT are transient. After the initial volume increase following ECT, the hippocampal volume gradually shrinks within 10–36 months after ECT has ended [77], though it may also occur as early as within the first 3 months [56]. As cognitive deficits after ECT are also transient, it is possible that changes in hippocampal volumes after ECT and cognitive impairment have some degree of relationship. In addition, impaired cognitive performance after ECT was also correlated with reduced resting-state functional connectivity (RSFC) of the hippocampal middle subregion with the bilateral angular gyrus [21]. Because the angular gyrus is involved in memory functions [87], RSFC reduction implies impaired connectivity between the hippocampus and the angular gyrus, which contributes to impaired cognition after ECT [21]. Taken together, these results showed a possible dual outcome of ECT-induced hippocampal volume increase, in which the mechanism necessary for the volume increase is also responsible for the cognitive deficit [84]. A possible explanation is that ECT is a potent stimulator of neurogenesis, which may increase the likelihood of causing aberrant or excessive neurogenesis that could disrupt hippocampal function [17]. The fact that both volume change and cognitive deficit are influenced by electrode placement and are temporary strengthens this argument.

## 3. ECT Induces Neurogenesis

### 3.1. ECS Promotes Hippocampal Neurogenesis in Animals

Structural neuroplasticity, especially pertaining to the hippocampus, has been explored in preclinical studies. Rats subjected to CRS displayed depressive behavior, but this was not reflected by any significant change in the rats’ hippocampal volume or in the total number of neurons [88]. In a later study, only the hilus showed a significant volume reduction [89]. The intervention of electroconvulsive stimulation (ECS), the animal equivalent of ECT, alleviated the depressive behavior. While ECS does increase neurogenesis in the hippocampal subgranular zone, it does not alter the total number of neurons or the volume of the hippocampus and its subdivisions [88,89], which may indicate damaging effects from ECS administration. On the other hand, ECS does stimulate neurogenesis, as indicated by the formation of new neuronal cells within the hippocampus [88,90].

The implied relationship between an increased number of ECT administrations and the magnitude of volumetric change in the hippocampus has been proven to occur on a cellular level. An increased number of ECSs given to rats produced more newborn neurons within the hippocampus [91]. However, ECS-induced neurogenesis was not fully efficient since there was a decline of around 40% of the newborn cells stained with the BrdU marker from day 1 post-ECS until 3 months, after which no further decline in newborn neuronal cells was observed for up to 12 months [90]. This early period of the first 3 months after ECS may be crucial for the survival of the newborn neuronal cells. Another study found that the number of cells stained with Ki-67 and NeuroD increased 1 day after ECS administration. However, after 8 days, the number of Ki-67-positive cells reduced while the number of NeuroD-positive cells increased [92]. Ki-67 is an intrinsic marker for recent cell proliferation which is expressed in all proliferating cells [93] and is used to identify the early stages of adult neurogenesis [94]. NeuroD is a transient transcription factor restricted to developing neurons that is essential for the survival and maturation of adult-born neurons [95].

ECS also has varying effects on the cell cycle. During the differentiation phase, ECS was found to increase cell survival, promote the differentiation of newborn cells, and significantly increase the expression of the immature neuronal marker calretinin. Conversely, ECS significantly suppressed the expression of the mature neuronal marker calbindin in newborn neurons during the late maturation phase [96]. It is apparent that ECS has a prominent impact on hippocampal neurogenesis. However, not all newborn neurons survive and mature before being incorporated into the functional hippocampal network. Hence, in order to improve neurogenesis potential, the effects and relationships between the various aspects of neuronal cell synthesis may be further explored.

Regarding the connectivity of new neurons formed after ECS, studies are rather sparse, even in animals. However, similar information could be extrapolated from status epilepticus (SE) studies. New neurons formed 2 weeks after SE was induced in rats were able to survive for up to 6 months and constitute a significant portion of the population of mature granule cells in the DG [97]. Although these neurons are integrated into the hippocampal neurocircuitry, they replace the pre-existing neurons rather than add to the population [97]. Neuronal degeneration was noted to occur in the DG, CA1, and CA3, alongside neurogenesis, after SE was electrically induced in rats in another study [98]. The electrophysiology of the new neurons formed after SE showed reduced excitatory and increased inhibitory synaptic input [98]. Additionally, neurons born after partial SE tend to have altered synaptic scaffolding proteins and adhesion molecules expression that favor the reduced excitability of these new neurons [99]. These changes could either reflect mechanisms to attenuate the epileptic dysfunction [98,99] or indicate immature inhibitory transmission due to the transient nature of elevated inhibitory molecules [99]. These findings could imply similar mechanisms occurring post-ECS and may also explain the cognitive outcome.

### 3.2. ECT Effects on BDNF Levels

The neurotrophic hypothesis of depression describes dysfunctional neuronal plasticity as the main mechanism underlying depression development. BDNF is a key component of this mechanism [100]. Rats subjected to unpredictable chronic mild stress (UCMS) display behavioral defects akin to those of depression, and this is accompanied by reduced BDNF concentrations in the CA1 and CA3 regions of the hippocampus [101] and also the PFC [102]. In a clinical setting, patients suffering from TRD displayed low serum BDNF levels [103,104]. Elderly patients with late-life depression displayed a similar reduction in serum BDNF that was significantly correlated with reduced hippocampal volume [105]. Cavaleri et al. noted that MDD patients have lower peripheral and central BDNF levels than non-depressed individuals, which have negative correlations to symptom severity but are not linked with suicidality [106].

Studies exploring serum BDNF changes after ECT produced varying outcomes, which may be partly due to the different methods of measuring peripheral BDNF levels. Serum and plasma BDNF have been explored as potential biomarkers in ECT studies. It was found that depressed patients who had undergone ECT displayed increased BDNF concentration in both their serum [107,108] and plasma [107]. However, serum BDNF is preferred over plasma BDNF. A study revealed no difference in baseline plasma BDNF levels between depressed patients and healthy controls [109]. The post-ECT readings either show no changes in concentrations [109,110] or display variations throughout an ECT course [111]. The tendency of plasma BDNF to display greater standard deviations in measurements [112], in addition to being influenced by the pre-analytical processing of blood samples [113], may lie in the manner in which BDNF is stored in serum and plasma [107]. Human BDNF is primarily stored in the platelets and is released mainly via platelet activation and degranulation during the clotting process. This explains why serum BDNF levels are higher than plasma levels [111]. Therefore, serum BDNF is the preferred method for peripheral BDNF measurement. Another recommendation is to employ serum-to-whole blood concentration of BDNF as a more accurate measurement of peripheral BDNF [114].

Even though increased serum BDNF after acute ECT [110] may indicate that ECT corrected the BDNF changes in depression, several studies have reported a lack of any significant changes in serum BDNF concentrations after ECT [104,105,111]. One study reported an insignificant trend of increasing BDNF [115], while another found that BDNF increased up to a certain value only in ECT non-remitters, with the end value resembling that of baseline remitters and no significant change in ECT remitters [116]. A recent overview of meta-analyses indicated that increases in blood BDNF levels can be seen after pharmacological treatment but are absent in non-pharmacological interventions, including ECT [106].

One possibility for the lack of change is that serum BDNF could have a delayed effect, requiring some time before any significant changes could be detected. According to a meta-analysis study by Luan et al., a significant increase in BDNF occurred one month after the completion of an ECT course [107]. Continuation ECT (c-ECT) is a course of less frequent ECT given after an index ECT course that can last up to 6 months [117]. The addition of c-ECT robustly increased serum BDNF concentrations [110,118], which may also support the idea of a serum BDNF increment time delay. However, the delayed increase in peripheral BDNF may not be attributed to the immediate clinical improvement after ECT, as the peripheral BDNF increase should have preceded the clinical response [104].

An alternative to both serum and plasma sampling is studying the cerebrospinal fluid (CSF). One study measured the BDNF concentration in CSF of MDD patients before and within a week after a course of ECT. The results indicate an increase in BDNF concentration in the CSF, which was absent in the corresponding serum samples [119]. However, the results are preliminary since the study is limited by its small sample size due to the exceptional difficulty in recruiting patients, especially for multiple CSF sampling [119]. Post-mortem sampling of brain tissue, CSF, and plasma from individuals with mood disorders and substance or alcohol abuse disorders indicated a significant positive correlation between BDNF in the brain and CSF, and between BDNF in the brain and plasma [120]. Considering BDNF changes in the serum [107], an increase in BDNF in the CSF may either occur concurrently with an increase in serum BDNF or it may slightly precede the changes in the serum. Nonetheless, future studies may explore the BDNF changes in CSF, serum, or plasma across different time points to validate the correlation.

The preclinical setting allowed for a better look at hippocampal BDNF levels. A study employed Flinders sensitive and resistant line rats, the genetic phenotypes that model depression. ECS significantly increased BDNF protein in both lines, and the increase was associated with a volumetric increase in the stratum radiatum in the CA1 regions in the sensitive line [121]. An earlier study found that ECS mediated the upregulation of BDNF protein in Wistar rats previously subjected to UCMS without altering the number of neurons in the CA1 and DG [122]. This effect was notably absent in the genetic variant [122]. While animal studies may provide a more accurate measurement of central BDNF, more studies are required in order to establish a direct relationship between hippocampal BDNF and volume increase.

### 3.3. Genetic Changes Influencing BDNF Expression as a Result of ECS

Although clinical studies are lacking, looking into animal models may shed some light on the ECS-mediated genetic changes influencing BDNF expression. Previous studies have established that ECS upregulated several immediate early genes (IEGs) such as *Bdnf*, *Egr1*, *c-Fos*, and *Nrn1* [123]. IEGs are genes that are rapidly activated in the brain in response to external stimuli and influence many neurobiological processes. One IEG, the early growth response 3 (*Egr3*) gene, was discovered as crucial for the expression of BDNF in the DG of the hippocampus in rats. The presence of several *Egr3* binding sites in the *Bdnf* promoter implies that *Egr3* directly mediates ECS-induced Bdnf gene expression, specifically exons IV and VI of the *Bdnf* gene [124]. Another IEG, *Narp*, was found to be an essential regulator for the ECS’s antidepressant effect. A study using *Narp* knockout mice discovered that *Narp* deletion inhibits the antidepressant effect of ECS in mice without blocking seizure activity. While the Narp protein was previously known to be secreted from pre-synaptic terminals to regulate AMPA receptor clustering and mediate BDNF-dependent synaptic modulation [125], *Narp* deletion neither suppressed hippocampal BDNF expression nor suppressed neurogenesis in the DG. However, it does block the dendritic outgrowth of the immature granule cell neurons. Therefore, although Narp protein does not influence ECS-induced hippocampal neurogenesis via BDNF synthesis, it may regulate the BDNF downstream effects on neuronal survival and maturation [126]. These discoveries add to the growing evidence that BDNF synthesis and the downstream pathway activities are regulated by a complex interplay of various genetic factors.

### 3.4. ECT Effects on BDNF Downstream Signaling

BDNF performs neurotrophic functions by binding to and phosphorylating its high-affinity receptor, tropomyosin receptor kinase B (TrkB) [127]. TrkB regulated hippocampal neural stem cell proliferation and neurogenesis in mice [128]. Multiple ECS performed in rats enhanced BDNF/TrkB signaling, evident by the increased expression of mature BDNF proteins as well as phosphorylated TrkB receptors in the dorsal and ventral regions of the hippocampus despite the reduced full-length TrkB receptor expression [129]. Increased TrkB signaling seems to occur during the resting period after intense epileptic activity has subsided [130]. In another study on mice, the period of slow EEG upon the discontinuation of N_2_O was discovered to have antidepressant effects with a concurrent increase in TrkB phosphorylation. This effect was also seen with the discontinuation of ketamine and flurothyl-induced convulsions, which mimic the effects of ECS [131].

Upon TrkB activation, BDNF causes the phosphorylation of protein kinase B (Akt) via the phosphatidylinositol 3-kinase (PI3K) pathway [127]. Both in vivo and in vitro studies have determined that PI3K-Akt signaling is involved in neuronal growth and survival since the upregulation of BDNF, TrkB, PI3K, and Akt proteins promotes axonal regeneration [132] and inhibits neuronal apoptosis [133,134]. While evidence connecting ECT and PI3K is currently lacking, it could be stipulated that ECT may promote PI3K upregulation as shown by its upstream and downstream factors, TrkB and Akt, respectively. In patients with TRD, there is reduced baseline expression of the microRNAs (miRNAs) let-7b and let-7c. Both miRNAs mediate the expression of genes in PI3K-Akt signaling, such as the gene for insulin growth factor 1 receptor (IGF-1R) [135]. IGF-1 is another neurotrophin besides BNDF, and it has been hypothesized that hippocampal BNDF upregulation is dependent on IGF-1 [136]. Although both miRNAs were unaffected by ECT [135], the evidence presented here suggests that depression may be influenced by PI3K-Akt activity.

In a manner similar to TrkB, there is a gradual upregulation of glycogen synthase kinase 3 beta (GSK3β) protein phosphorylation during the rebound of slow brain EEG oscillations upon the discontinuation of the antidepressants ketamine and N_2_O to produce their antidepressant effects [131]. This phenomenon implies a similar outcome in ECS and possibly ECT upon subsidence of their electroconvulsive activity. GSK3β activity is inhibited upon its phosphorylation by Akt and protein kinase C (PKC). GSK3β itself inhibits Wnt/β-catenin signaling by promoting β-catenin degradation via phosphorylation [137]. Wnt/β-catenin signaling is crucial for hippocampal neurogenesis [138] and protection from neuronal apoptosis [139]. The microRNA mIR-155 induced depression-like behaviors in mice, possibly via the inhibition of Wnt/β-catenin signaling [139]. ECS in mice is proven to affect multiple components of a complex pathway, as ECS promotes Akt activation and PKC and Wnt/β-catenin signaling, besides reducing GSK3β activity by increasing its phosphorylated form [140].

Besides Akt signaling, BDNF is also the immediate upstream regulator of extracellular-signal-regulated kinase 1/2 (ERK1/2) signaling. Reduced ERK1/2 signaling was found in the PFC and hippocampus of both human and animal depression models [141]. ERK1/2 is involved in several neurobiological processes, such as promoting neurogenesis and synaptogenesis in the early stages of life and long-term potentiation in adulthood [142]. Elevated levels of phosphorylated ERK1/2 (p-ERK1/2) and ERK1/2 have been associated with the amelioration of depressive symptoms and improved cognitive performance in rats previously subjected to UCMS [143]. Repeated ECS administration in rats causes an ECS-induced cognitive deficit, which is associated with reduced p-ERK1/2 levels and excessive activation of the N-methyl D-aspartate (NMDA) receptor subunit NR2B [144]. Both ERK1/2 and NR2B seem to be influenced by the number of ECS given. A single, but not repeated, ECS was able to induce a transient increase in p-ERK1/2, and this increase occurred without involving the activation of its upstream factor TrkB [145]. The possible reason given is that a single ECS induces an acute membrane depolarization in the PFC and this eventually leads to the activation of ERK1/2 signaling via increased NMDA activity [145]. In contrast, repeated ECS has been shown to reduce NMDA receptor 1 and increase NR2, with the former being associated with cognitive deficit while the latter is associated with improved long-term memory by neurogenesis [140].

The role of NR2B is somewhat muddled by the fact that increased NR2B activity has been linked to both cognitive dysfunction [146] and improvement [147,148]. For cognitive improvement, the administration of piperlongumine, a compound derived from *Piper* sp., to aged female C57BL/6J mice improved their cognitive performance. The improvement was attributed to the increased phosphorylation of NR2B, ERK1/2, and CamKIIα in the mouse hippocampus, as well as increased neurogenesis in the DG [147]. Another study used a rat model of hepatic ischemia/reperfusion that displayed cognitive dysfunction features and reduced mRNA and protein expression of NR2B in the hippocampus. Treatment with hydrogen sulfide improved their performance on the Morris water maze test while also increasing the NR2B protein levels [148]. On the other hand, one study exposed rats to the anesthetic isoflurane, which increased calpain-2 expression in the hippocampus secondary to increased NR2B expression [146]. Calpain is a Ca^2+^-activated neural cellular cysteine protease, and it has the isoforms calpain-1 and calpain-2. While calpain-1 is neuroprotective, calpain-2 has neurodegenerative effects [149]. The increased expression of calpain-2 reduced the potassium-chloride cotransporter (KCC2) and total 4.1N protein expression [146]. Both KCC2 and 4.1N proteins, the latter being a cytoskeletal-associated protein, are significant in the formation of dendritic spines [150], and their suppression results in spatial memory deficits in rats [146]. In turn, NR2B inhibition reduced calpain-2 expression and increased KCC2 and 4.1N protein in the rats, alleviating the spatial memory deficit [146]. NR2B may have multiple pathways leading to both outcomes, which may also be affected by other neurogenic pathways. No study thus far has linked ECT and calpain-2, so it is interesting to see whether calpain-2 has a role in the cognitive side effects post-ECT. A diagram of the effects of ECC on BDNF signaling, based on the results of preclinical studies, is shown in Figure 1.

### 3.5. Potential Role of proBDNF in Disrupting Cognitive Function Following ECT

Mature BDNF arises from the proteolytic cleavage of its isoform precursor, proBDNF [151], a process regulated by tissue plasminogen activator (tPA) [152]. While mature BDNF promotes neurotrophic effects by binding to the TrkB receptor, proBDNF depresses synaptic plasticity and induces neuronal apoptosis by binding to the neutrophin receptor p75NTR [153]. The protein p75 induces the internalization of the intracellular domain (ICD) into the nucleus, resulting in cell apoptosis [154]. Clinical studies have shown that serum BDNF and tPA levels were lower in depressed patients, while serum proBDNF and p75NTR levels were higher [155,156]. Another study reported that plasma BDNF levels were significantly reduced in patients with BD compared to MDD patients, and that MDD patients had a higher ratio of mature BDNF to proBDNF in the plasma than BD patients or healthy controls. This indicates the mature BDNF to proBDNF ratio may be a differentiating factor between MDD and BD [157].

In preclinical studies, mice exposed to UCMS displayed increased protein expression of proBDNF and p75NTR in the neocortex and hippocampus [158,159]. The chronic administration of corticosterone in mice induced depressive-like behaviors, which are accompanied by an elevated proBDNF level in the ventral DG and a reduced BDNF level in the dorsal and ventral DG [160]. Exogenous proBDNF administration was also found to induce depression-like symptoms. Lin et al. employed an adeno-associated virus (AAV) vector to deliver proBDNF via injection into the bilateral gluteus maximus in mice. The AAV was able to induce depression-like behaviors in mice after about 4 weeks, and the administration of sheep anti-proBDNF antibody was able to alleviate these symptoms [161]. The direct injection of exogenous proBDNF into the hippocampal CA1 in rats via an implanted steel cannula produced depressive-like symptoms, similar to the effect of UCMS [162]. On a cellular level, the exogenous delivery of proBDNF reduced the density of dendritic spines in the hippocampal CA1 region [162], the DG, and also the amygdala [161]. Finally, as proBDNF and its receptor are expressed in neural stem cells, exogenous proBDNF treatment reduced the neural stem cells’ proliferation, migration, and overall viability [163]. Thus, it can be deduced that proBDNF may be a potential mediator in the development of depression symptoms.

ECS in rats was shown to upregulate the expression of mature BDNF and TrkB receptors without altering the overall proBDNF expression in both the dorsal and ventral hippocampus [129]. However, an earlier study demonstrated an upregulation of proBDNF in the CA1 regions following the ECS of depressed rats, alongside increased mature BDNF and tPA levels, as well as an increased proBDNF/mature BDNF ratio. The levels of proBDNF and mature BDNF were previously elevated and reduced, respectively, in these rats after being subjected to UCMS, which may be due to tPA inhibition secondary to increased expression of plasminogen activator inhibitor-1 (PAI-1), as there are increased PAI-1 levels after UCMS [152]. It has been suggested that elevated proBDNF in the CA1 regions causes impaired long-term potentiation and may be responsible for the cognitive deficit following ECT [152]. Increased proBDNF levels after UCMS were associated with impaired cognitive ability, whereas ECS was associated with a further increase in proBDNF levels and a worsening cognitive deficit [152]. Elevated proBDNF and reduced mature BDNF have been associated with altered cognitive functioning in mice following radiation therapy, in which irradiated mice engaged in a cognitive task at a slower pace [164]. However, the irradiated mice did not show any reduction in terms of their engagement with the cognitive task, implying that fatigue after irradiation could also influence the cognitive behaviors [164]. Diminishing cognitive ability, as seen in Alzheimer’s disease (AD), has been associated with elevated proBDNF and p75 activity [154]. CSF from AD patients was proven to induce apoptosis in cultured hippocampal neurons, and this effect was mediated by proBDNF found in the CSF [154]. Taken together, the role of proBDNF can be investigated further. While it is clear that proBDNF plays a role in depression, both proBDNF and mature BDNF were also increased after ECS in rats. The exact mechanism by which both proBDNF and mature BDNF increase concurrently after ECS is unknown, but other mechanisms besides the tPA system might be involved in temporarily elevating proBDNF after ECT. It could also explain why proBDNF is elevated in both depression and after ECT. Further studies are warranted to solidify the role of proBDNF-p75 signaling in cognitive deficit, as well as to determine whether there is a temporal change in this signaling that coincides with the temporal changes in cognitive ability following ECT.

## 4. VEGF Changes after ECT

Another growth factor of interest besides BDNF is the vascular endothelial growth factor (VEGF). Originally known as an angiogenic factor, VEGF also exhibited neurotrophic properties. It has been linked to depression since patients given anti-VEGF medications exhibited depression and anxiety behaviors [165,166]. In an animal study, VEGF treatment was able to ameliorate depression-like behavior in mice [167]. According to a review by Deyama and Duman, ketamine, an NMDA receptor antagonist, produced robust antidepressant effects that were associated with increased BDNF and VEGF expression in the medial PFC and hippocampus [168]. However, clinical studies have produced inconsistent results. Some studies reported no difference in blood VEGF levels before ECT between depressed patients and healthy controls [169,170], while another found lower peripheral VEGF in TRD patients compared to healthy controls [103]. A similar finding was also found in the CSF of TRD patients [171]. On the other hand, several meta-analysis studies pointed out that blood VEGF levels in MDD patients were significantly higher compared to healthy controls [172,173,174]. Higher baseline VEGF levels were also found in treatment-resistant bipolar depression patients [175]. A plausible explanation for the difference in VEGF levels between TRD and non-TRD is that higher VEGF levels in non-TRD are reflective of the brain’s neuroprotective mechanism in response to stress. This mechanism is absent in TRD patients, preventing antidepressant medication response [103]. This is evident from lower VEGF levels that were also seen in treatment-resistant schizophrenia patients; they were increased following ECT [170,176]. Plasma VEGF samples taken 1–3 days after ECT treatment were shown to be unaltered [169]. However, a later study showed that there was an increment in plasma VEGF level when taken 2 h after ECT, and the level decreased when taken 4 h post-ECT [177]. Hence, it could be deduced from here that increased VEGF levels after ECT might be temporary, but VEGF has some degree of involvement in the mechanism of ECT.

Among the VEGF mammalian subtypes, VEGF-A is of particular interest. VEGF-A induces vascular changes such as promoting angiogenesis, vasodilation, and increased vascular permeability. It also possesses neurotrophic actions in that it induces neurogenesis via its receptor VEGFR2 and also promotes neuronal survival via the PI3K-Akt pathway and the mitogen-activated protein kinase (MAPK) pathway [178]. While VEGF is expressed in different cell populations in the hippocampus, VEGFR2 is specifically expressed in the pyramidal neurons of the CA3 subfield, as revealed in an animal study [179]. VEGF-A/VEGFR2 signaling also promoted axonal branching in the rats’ hippocampal neurons [179]. Another study revealed that VEGF-A/VEGFR2 signaling mediates neural stem cell proliferation and migration in immature rats after SE [180]. The increased expression of the VEGF-A downstream proteins VEGFR2 and phosphorylated Akt in the hippocampus following SE supports this [180]. Thus far, there have been no studies investigating the effects of ECS on VEGF-A. The aforementioned findings in animal studies contradict a recent clinical study. Peripheral mRNA expression of VEGF-A was found to be higher in patients with psychotic and non-psychotic depression compared to healthy controls, which was then reduced following ECT [181]. However, the small sample size, as well as ongoing medications that were taken by patients during the study, may confound this outcome [181]. Regardless, VEGF-A is a promising target for future ECT-related studies. Figure 1 depicts the role of VEGF-A in neurogenesis following ECT.

Additionally, it is worth mentioning the study by Van Den Bossche et al. regarding genetic polymorphisms influencing changes in hippocampal volume after ECT. They noted that the expression of VEGF can be regulated by certain single-nucleotide polymorphisms (SNPs), particularly rs699947 in the promoter region of VEGF. It is possible that more C alleles at rs699947 lead to higher VEGF expression following ECT, resulting in a higher increase in hippocampal volume. The association between SNP and hippocampal volume change is not found in rs6265 for BDNF [182]. Genetic studies may provide a further understanding of the role of VEGF in the ECT response.

## 5. ECT and Cerebral Edema

### 5.1. Compromised Neuronal Integrity Post-ECT

According to diffusion tensor imaging (DTI) studies, white matter structure can be affected by ECT. A reduction in hippocampal fractional anisotropy (FA) occurred a week after ECT [115], and is associated with seizure duration [183]. As FA is an index measurement of white matter integrity [184], a reduction in FA may indicate compromised white matter integrity as a result of ECT. A lower FA was also found in depressed patients with childhood trauma [185], as well as reduced FA in the peripheral neurons of patients with spinal cord injury [186]. Both indicate neuronal dysfunction with reduced FA, and a reduction in FA following ECT could indicate potential cognitive dysfunction due to compromised neuronal integrity. In contrast, other studies showed no significant changes in FA following ECT [57,187]. However, FA is not always a reliable marker of white matter integrity [184], as it is a non-specific measure of diffusion. Other diffusion metrics, such as mean diffusivity (MD), axial diffusivity (AD), and radial diffusivity (RD), can change without significantly affecting FA [57].

Another marker of neuronal integrity is N-acetyl aspartate (NAA), obtained from proton magnetic resonance spectroscopy (H-MRS) imaging. Reduced NAA levels were found in the left hippocampus of depressed patients [188] and in youths with TRD [189], as well as in the posterior cingulate cortex in elderly patients with late-life depression [190]. Post-ECT measurements indicate that these changes were not corrected with ECT, as evidenced by reduced NAA concentrations in the left medial temporal lobe [76], the dorsal anterior cingulate cortex [188], and the PFC and orbitofrontal cortex [191] after ECT treatment. Interestingly, NAA was also decreased in the right hippocampus following RUL ECT, which occurs in depressed patients with an already-reduced NAA in the left hippocampus [188]. In a recent review, it was revealed that NAA reduction provided the strongest evidence of neurochemical alterations with ECT [192]. This evidence further supports the notion of disturbed neuronal integrity following ECT. It is possible that NAA reduction is a predominant factor in the stimulation of subsequent neurotrophic processes [188] or that it reflects a transient neuronal loss that leads to cognitive side effects due to PFC dysfunction [191]. In patients with bipolar type II depression, reduced NAA relative to creatinine ratios in the bilateral prefrontal white matter were associated with impaired cognitive performance [193].

### 5.2. Could ECT Result in Cerebral Edema?

Measurements of other DTI indices post-ECT point to a predisposition for increased neuronal diffusivity in the right hemisphere [57,183,187], which could be due to the utilization of RUL ECT [187]. RUL ECT increased MD, AD, and RD in the right hippocampus, and this effect was specific to treatment responders only [57]. ECT-induced reduced white matter integrity may reflect increased permeability of the blood–brain barrier (BBB) [183], suggesting a potential glial or inflammatory response mechanism affecting neuronal integrity [57], which drives fluid shift into the cerebral interstitium. However, the notion of post-ECT cerebral edema is unsupported by other studies. Imaging studies with the fluid attenuated inversion recovery (FLAIR) sequence indicated no edematous changes in the hippocampus post-ECT [194,195]. Another study found a significant decrease in MD in the bilateral hippocampus after ECT, which was attributed to the extracellular space in the hippocampus becoming more occupied due to neurogenesis or synaptogenesis, leaving less space for free water diffusion [196]. It was also determined that gray matter volume changes mainly drove the effects of ECT on the medial temporal lobe, while myelin changes and tissue free water drove the effects in the ventromedial PFC and the anterior cingulate cortex [197]. These findings suggest that neither vasogenic edema nor angiogenesis is responsible for the ECT-induced volume increase in the hippocampus [196,197].

### 5.3. ECT Induces Cytotoxic Edema in Astrocytes by Way of Aquaporin and Glutamate

Astrocytes are a group of glial cells that support the neuronal cells in the central nervous system (CNS). Astrocytes are involved in homeostatic functions, neuroprotective mechanisms, and synaptic modulation [198]. They are linked to neurotrophic processes by increasing leukemia inhibitory factor (LIF) expression after receiving electrical stimulation, in which LIF stimulates Wnt signaling [199]. Other astrocyte-derived factors such as VEGF and glutamate enhance the permeability of the BBB [200]. Astrocytes possess endfeet processes that ensheath blood vessels in the brain to form gliovascular units that integrate the neural circuitry with local blood flow [201]. Following an acute insult, cytotoxic edema occurs in all CNS cell types but is especially prevalent in astrocytes [202].

ECS in rats transiently enhanced the permeability of the BBB together with swelling in the astrocyte endfeet [201]. A study with Gunn rats, a phenotypic model of affective disorders, showed that ECS improved the depressive-like behavior in the rats. This is accompanied by increased astrocytic blood vessel coverage, which was previously reduced in the model, and also elevated aquaporin-4 (AQP4) expression [203]. These post-ECS changes are not accompanied by any disruption in the tight junctions [203]. Gunn rats also exhibited elevated astrocyte activation in the hippocampus, as evidenced by increased immunoreactivity for glial fibrillary acidic protein (GFAP), which was attenuated with ECS [204]. However, a later review ascertained that chronic ECS induced the transient activation of inflammation and a longer-lasting activation of neurogenesis and astrocytes, as indicated by a prolonged elevation of GFAP for astrocyte activation [205]. Glial activation occurs alongside neurogenesis and temporary inflammation, all of which may partake in the therapeutic effect of ECT.

AQP4 is the main water channel protein in the CNS and is densely expressed in astrocyte endfeet [206]. Increased AQP4 activity, which causes neuronal swelling, may be dependent on the glutamate level in the brain. Glutamate acts on the metabotropic glutamate receptor 5 (mGlu5), whose expression was highest among the metabotropic glutamate receptors in cultured astrocytes and was found to be co-expressed with AQP4 [207]. Glutamate increased AQP4 expression and caused neuronal swelling by binding to mGLu5 [207]. Reduced FA, indicative of reduced neuronal integrity, is also associated with an increased number of swollen astrocytes, immunophenotyped by AQP4, GFAP, and the glucose transporter GLT-1 [208]. GLT-1 is responsible for transporting glutamate into astrocytes before it is converted into glutamine [209]. GLT-1 deficiency has been linked to temporal lobe epilepsy, which promotes an increase in glutamate [209]. Given that AQP4 is expressed within the DG and CA1 of the hippocampus [210], it could be implied that a similar mechanism exists in ECT. The epileptic state induced by ECT may lead to an enhanced activity of glutamate on AQP4, increasing BBB permeability and resulting in edema in the neuronal cells, which may account for the transient cognitive loss post-ECT.

After H-MRS analysis of depressed patients, researchers discovered a reduction in glutamate and glutamine (Glx) concentrations in the subgenual anterior cingulate cortex [188] and the medial frontal cortex [211], as well as an increase in the left hippocampus [188]. Elevated Glx levels in the right hippocampus were also found in adolescents and young adults diagnosed with TRD [189]. Both the subgenual anterior cingulate cortex and the medial PFC have been implicated in the pathophysiology of depression [212,213]. Following ECT, the outcomes varied. Some studies reported increased Glx concentrations after ECT [76,192], while others found no difference between pre- and post-ECT concentrations in the PFC, OFC [191], and the anterior cingulate cortex [214]. The effectiveness of ketamine also contradicts glutamate’s role in mediating cognitive decline. Ketamine, an NMDA receptor antagonist, opposes the action of glutamate and is thought to provide some alleviation of cognitive impairment. When used alone or combined with haloperidol, however, ketamine administration was unable to prevent cognitive decline either in patients undergoing an ECT course or in patients undergoing surgery [215,216]. However, it is possible that the role of glutamate in mediating cognitive decline is due to its action on glial cells rather than synaptic processes. Nevertheless, in a scenario without a reduced baseline Glx in depressed patients, an elevated Glx post-ECT may reflect the inflammatory process leading to neuronal swelling [76,217].

### 5.4. The Potential Role of mGlu5 and Homer1a

Homer1 is a scaffolding protein that mediates metabotropic glutamate receptors [218,219], including mGlu1 and mGlu5, signaling in neurons. Both stress and ECS administration in mice were noted to induce upregulation of Homer1 expression, which was attributed to the prevention of stress-induced dendritic changes in the hippocampal CA3c region [220]. Homer1a has been implicated in antidepressant mechanisms. In another preclinical study, Homer1a was fused to the protein transduction domain (TAT) to allow cell permeability and delivery of the protein to the brain through the BBB. The peripheral administration of TAT-Homer1a in mice to resemble Homer1a upregulation produced antidepressant effects via enhanced mGlu5/mTOR signaling, resulting in enhanced AMPA receptor expression [221]. Increased Homer1a expression by reactive astrocytes reduced the intensity and duration of glutamate release by astrocytes secondary to a reduction in Ca^2+^ signaling [222]. Given that mGlu5 expression in astrocytes was accompanied by AQP4 expression [207], both the mGlu5 receptor and the Homer1a protein may be potentially involved in the mechanism of cognitive decline post-ECT. One study reported that mGLu5 receptor antagonists improved cognitive performance. Mice with a genetic model similar to human phenylketonuria (PKU) had higher mGLu5 receptor protein levels in the hippocampus, and they also displayed behavioral impairments in object exploration and spatial memory. The administration of 2-methyl-6-phenylethynyl-pyridine hydrochloride (MPEP), a mGlu5 receptor antagonist [223], significantly ameliorated the behavioral impairments and improved the mice’s cognitive performance [224]. Hence, further investigations are warranted to delve into the potential correlation between mGLu5 receptor and Homer1a expression and clinical and cognitive outcomes after ECT.

## 6. Conclusions

While it is clear that ECT does indeed cause volume changes in the hippocampus, the exact mechanism that underpins these changes still needs further elucidation. It is most likely that the mechanism for volume changes is also responsible for causing the cognitive side effects, as both the volume increase and the cognitive deficit are temporary and are influenced by electrode placement. ECS in animal studies has been shown to promote both neurogenesis and neuronal cell death (Figure 1). One possibility is that BDNF and VEGF mediate ECT-induced hippocampal neurogenesis, which helps to allay depressive symptoms while also causing excessive neurogenesis in other areas such as the PFC, disturbing the hippocampal cognitive function. Besides BDNF, ECT may also stimulate proBDNF and NR2B signaling that leads to neuronal cell death and degeneration, disturbing the cognitive network. Besides neurogenesis, we also evaluate the potential role of edema in mediating cognitive deficits post-ECT. While FLAIR imaging may have ruled out vasogenic or ionic edema, we cannot ignore evidence showing increased diffusivity in the white matter. Therefore, we theorize that ECT may cause astrocyte swelling that is transient enough not to cause ionic edema but widespread and prevalent enough to impede signal transmission and disturb neural function. As shown in Figure 2, this effect could be mediated by AQP4 and mGLu5, as well as the protein Homer1a. Although this theory is based on current results from animal studies, it may help direct future studies with ECT, especially in regard to improving the cognitive outcome.

## Figures and Tables

**Figure 1 diagnostics-13-01585-f001:**
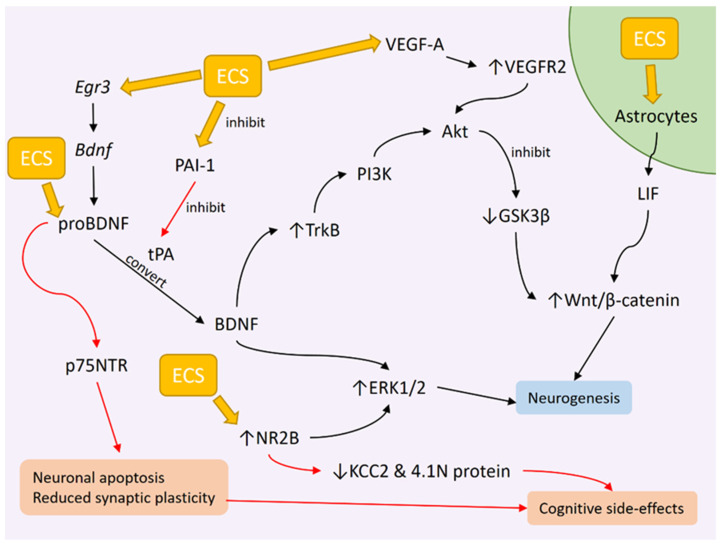
The mechanism of ECS, based on preclinical studies. Black arrows indicate pathways leading to neurogenesis. Red arrows indicate pathways leading to cognitive side effects.

**Figure 2 diagnostics-13-01585-f002:**
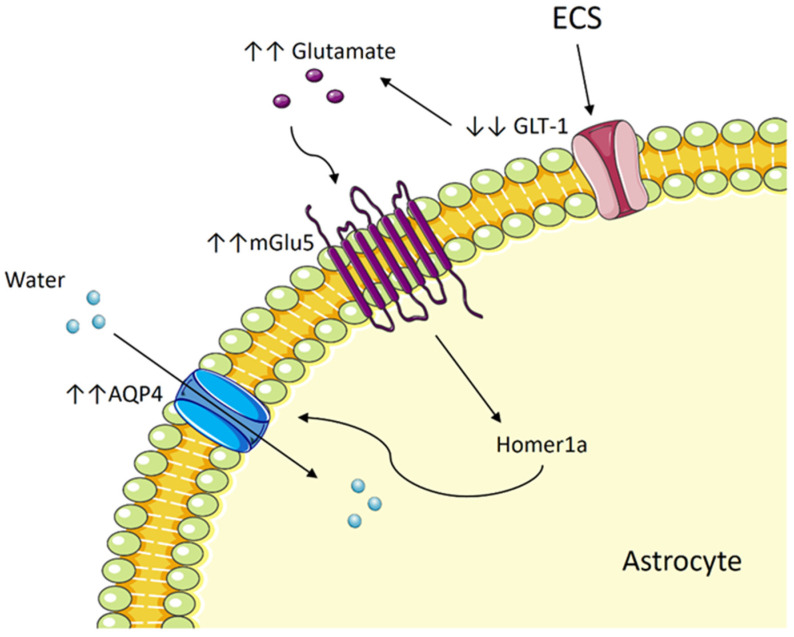
The proposed mechanism for how ECT causes transient cytotoxic edema in astrocytes, based on current findings from in vitro and animal studies.

**Table 1 diagnostics-13-01585-t001:** Brain changes following ECT that are significantly correlated with cognitive impairments.

ECT Electrode Placement	Brain Changes	Cognitive Outcome	Reference
Bifrontal	Increased cortical thickness in the left inferior parietal gyrus and reduced surface area of right inferior temporal gyrus	Impaired delayed memory	[83]
Bitemporal	Increased volume in bilateral hippocampus	Slower processing speed and impaired divided attention ability, impaired delayed memory	[84]
Bitemporal	(Immediately after ECT series) increased volume in right and left DG	Reduced delayed memory performance	[17]
	(At 6-month follow-up) reduced volume in right DG	Improved delayed memory performance	
RUL	Larger absolute increase in right hippocampal volume	(Assessment at 6 months) less improvement in visual memory	[85]
Switched to BT after failed RUL	Larger absolute increase in right hippocampal volume	(Assessment at 6 months) less improvement in semantic memory and verbal memory, less improvement in global cognitive functioning	
Bifrontal	Reduced RSFC between left hippocampal middle (cognitive) subregion and bilateral angular gyrus	Impaired performance on the verbal fluency test	[21]
Bifrontal (for major depressive episode) and bitemporal (schizophrenia spectrum disorder)	Increased volume in the hippocampus and amygdala	Lower Repeatable Battery of the Assessment of Neuropsychological Status (RBANS) scoring	[86]

## Data Availability

Not applicable.

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
