# Peer review of "Clinical Improvement in Depression and Cognitive Deficit Following Electroconvulsive Therapy"

_diagnostics, 2023, doi:10.3390/diagnostics13091585_

Round 1

Reviewer 1 Report

The review under consideration is the valuable impact in the knowlege of clinic investigators and experimental neurobiologists. It covers many dozens of references and will be  of use during long time.The data described  concern hippocampal volume changes (with details concerning BDNF  et al. moleciles) and  cognitive  issues connected to electric seizures induction. The overall conclusion that the whole problem needs further investigation is valid and authors impact (by this text) is the serious impact.

The test is fine and to my mind need probably minor changes.

The review under consideration is the valuable impact in the area of cognitive changes after ECT needed for clinic investigators and experimental neurobiologists. It covers many dozens of references and will be of use during long time. The data described hippocampal volume changes (with details concerning BDNF  et al. molecules) and  cognitive  issues connected to electric seizures induction. The overall conclusion that the whole problem needs further investigation is valid and authors’ impact (by this text) is the serious one.

The minor recommendations to authors are the following.

I failed to find any data concerniong the type of seizures (and their description) in patients.  It is worthwhile to make the short description as there will be neurobiologists among readers who don't know these clinic details.

In section 2.5 it would be good to give the refs concerning behavioral changes after ECT in animal models.

It would be good for readers to indicate, which refs concern patients and which are  the animal data (in general it was done, but not in several cases)

It also is worth mentioning that (at least in animal studies) the data concerning the function of new neurons (their connectivity) are sparse There are several papers (I was not able to find them in the refs list) that could be of use in this respect.
Jakubs et al. Environment matters: synaptic properties of neurons born in the epileptic adult brain develop to reduce excitability.
Neuron. 2006 Dec 21;52(6):1047-59. doi: 10.1016/j.neuron.2006.11.004.PMID: 17178407

Bonde et al. Long-term neuronal replacement in adult rat hippocampus after status epilepticus despite chronic inflammation. Eur J Neurosci. 2006 Feb;23(4):965-74. doi: 10.1111/j.1460-9568.2006.04635.x

Jackson et al. Altered Synaptic Properties During Integration of Adult-Born Hippocampal Neurons Following a Seizure Insult. PLoS One. 2012; 7(4): e35557. PMID: 22539981

Reviewer 2 Report

The Authors conducted massive work to synthesize all available evidence on this relevant topic. The article represents a commendable effort and is overall well written. 

Nonetheless, I have some comments that can further improve the contents and the value of their review. Please find my suggestions hereafter. 

·      In the Introduction section, at lines 72-75, the Authors refer to “the current state of knowledge”. Here they should mention a recent commentary by Cavaleri and Bartoli, 2022 [https://10.5152/alphapsychiatry.2022.0003] comprehensively discussing the state of the art and relevant limits of the available evidence on biomolecular mechanisms putatively underlying the biological response to ECT and its implications in terms of both efficacy and adverse effects, including cognitive side effects and their mechanisms.

·      In subsection 2.1, the Authors should consider evidence supporting the hypothesis that hippocampus enlargement is associated with worse cognitive outcomes, a finding that seems generalizable across different diagnoses, different electrode placements, and a different number of ECT sessions [https://doi.org/10.1038/s41398-021-01641-y]

·      In subsection 3.2, the Authors should mention the findings of a very recent overview on, amongst others, the effect of various treatments on blood BDNF concentrations in Major Depressive Disorder [https://doi.org/10.1016/j.neubiorev.2023.105159]

·      Regarding polymorphisms of genes that have been hypothesized to influence the changes in hippocampal volume following ECT – i.e., rs699947 (in the promotor region of VEGF) and rs6265 (in BDNF) – it was found that rs699947 had an effect on hippocampal volume changes following ECT, while rs6265 did not have a clear effect [https://doi.org/10.1038/s41398-019-0530-6].

Round 2

Reviewer 2 Report

No further comments.